# Manufacturing of Al_2_O_3_/Ni/Ti Composites Enhanced by Intermetallic Phases

**DOI:** 10.3390/ma14133510

**Published:** 2021-06-24

**Authors:** Marcin Wachowski, Justyna Zygmuntowicz, Robert Kosturek, Katarzyna Konopka, Waldemar Kaszuwara

**Affiliations:** 1Faculty of Mechanical Engineering, Military University of Technology in Warsaw, gen. Sylwestra Kaliskiego 2 St., 00-908 Warsaw, Poland; marcin.wachowski@wat.edu.pl (M.W.); robert.kosturek@wat.edu.pl (R.K.); 2Faculty of Materials Science and Engineering, Warsaw University of Technology, 141 Woloska St., 02-507 Warsaw, Poland; Katarzyna.Konopka@pw.edu.pl (K.K.); Waldemar.Kaszuwara@pw.edu.pl (W.K.)

**Keywords:** slip casting, composites, Al_2_O_3_/Ni/Ti, intermetallic phases

## Abstract

In this study, ceramic–metal composites in the Al_2_O_3_/Ti/Ni system were fabricated using the slip casting method. Two series of composites with 15 vol.% metal content and different solid phase contents were obtained and examined. A proper fabrication process allows obtaining composites enhanced by intermetallic phases. The microstructure of the base powders, slurries, and sintered composites was analyzed by scanning electron microscope. Analysis of the sedimentation tendency of slurries was carried out. The phase composition of the sintered samples was examined by X-ray diffraction analysis. A monotonic compression test was used to investigate the mechanical properties of the composites. A fractography investigation was also carried out. The research conducted revealed that the slip casting method allows the obtaining of composites enhanced by intermetallic phases (TiNi, Ni_3_Ti). The results show the correlation between solid-phase content, microstructure, and mechanical properties of the composites.

## 1. Introduction

One of the fastest growing fields of mechanical and materials engineering is research on the fabrication of innovative ceramic materials. Among these materials are ceramic–metal matrix composites. Ceramic–metal composites belong to the group of structural and functional engineering materials. Manufacturing such a material makes it possible to produce composites with different physical and chemical properties than a single-component material. This is particularly advantageous for materials with applications such as e.g., thermal barriers. Ceramic–metal composites are characterized by high corrosion resistance at high temperatures, thermal resistance, and high mechanical strength. The addition of a metallic phase into the ceramic matrix positively increases the fracture toughness. Although ceramic–metal composites have already been widely applied to various applications, the knowledge of the technological processes of their forming and a basic description of the relationships between their structure and properties is still insufficient. Ceramic–metal composites have aroused the interest of the world of science for years, thanks to their unusual properties, which are the sum of the beneficial features of the components [1,2,3]. A special type of ceramic–metal composite is the three-component material. The selection of appropriate metallic components allows the sintering process to create new phases [2,3,4], including intermetallic ones [5,6,7], thereby obtaining unique properties of the composite [7,8,9]. Taking the advantage of the properties of the individual materials (e.g., the electric conductivity and ductility of metal and the hardness and heat-resistance of ceramics) has made ceramic–metal composites very promising materials for use in biomedical [10], aerospace [11], electrotechnics [12], and other engineering applications [13,14].

An innovative composite in which it is possible to obtain intermetallic phases is the Al_2_O_3_/Ti/Ni composite produced by slip casting. The use of titanium and nickel as additional metallic components allowed the sintering process to produce intermetallic phases in the composite, which significantly affect the properties of the composite. Currently, forming methods based on colloidal processes play a leading role in the technology of advanced ceramic-based composites, including tape casting and direct coagulation casting [15], slip casting [16], gel casting [17], and centrifugal slip casting [18,19]. The results of our own research and data from the literature on using the slip casting method for the production of ceramic–metal composites show that it is one of the most effective methods of forming such composites [20,21,22,23,24]. In recent years, intermetallic compounds have often been used to improve the mechanical properties of metallic materials, including light-alloy multilayer composites [15,16,17,18,19,20,21,22,23,24,25,26,27].

Regarding ceramics, intermetallic-based reinforcement is a relatively new field of study with great potential for materials operating in extreme environments (e.g., high wear and high temperature). Klassen et al. performed research on the properties of intermetallic–ceramic composites. They found that Al_2_O_3_ reinforced with intermetallic compounds shows better damage tolerance (bending strength and fracture toughness) compared to monolithic Al_2_O_3_, combined with high wear resistance and electrical conductivity [28]. Wu et al. investigated Si_3_N_4_ ceramic joints with formed in situ Ni/Ti intermetallic compounds in terms of performance in high-temperature conditions [7]. They discovered that the formed Ni_3_Ti layer significantly increases the shear strength of the composite [8].

The Ni–Ti phase diagram describes three stable intermetallic phases: NiTi, Ni3Ti, and NiTi2, characterized by high hardness, wear resistance, and good mechanical properties at high temperatures [29,30,31]. The use of nickel and titanium as metallic components in the sintering process allows the obtaining of Ni–Ti intermetallic phases, which affect the final properties of the ceramic matrix composite [28,32,33,34,35]. However, there are no reports on this type of material in the subject literature, which is a scientific gap. Therefore, taking up the possibility of obtaining Al_2_O_3_/Ti/Ni composites by slip casting contributes to the knowledge of ceramic–metal materials reinforced with intermetallic phases. This research aims to examine the possibility of manufacturing by slip casting and to characterize the microstructure and mechanical properties of Al_2_O_3_/Ti/Ni composites reinforced with intermetallic phases. The results will allow for the basic development of a methodology for influencing the properties of composites with a ceramic matrix by introducing a metallic phase, which in the sintering process will form an intermetallic reinforcement and be an important reference point for future work on the possibility of obtaining three-component gradient ceramic–metal composite materials. The determination of the relationships between the content of the composite’s components, microstructure (including the formation of intermetallic phases), and mechanical properties is a new contribution in the field of engineering manufacturing of new composite materials and the results of this study are presented in the paper.

## 2. Materials and Methods

### 2.1. Base Powders

In the present study, the powders used to prepare the slurries and sintered composites were aluminum oxide TM-DAR (Tamei Chemicals, Tokyo, Japan). Used as a composite matrix, and the metallic powders: titanium (Alfa Aesar, Haverhill, MA, USA), and nickel (Alfa Aesar, Haverhill, MA, USA).

The choice of materials used to fabricate the composites is not accidental. The choice of the Al_2_O_3_ (TM-DAR) powder was mainly determined by their very high purity and submicrometric particle size. In addition, these powders are regularly used by investigators, thus providing for comparisons to be made with international study results. Furthermore, the aluminum oxide (TM-DAR) was used as a matrix due to its commercial availability. On the other hand, the choice of Ni and Ti powders was dictated by the fact that they easily form intermetallic phases. Moreover, the chosen metal powders are characterized by good thermal and ductility, electrical conductivity, and corrosion resistance.

In the first stage of the research, the base powders were investigated. Laser analysis of particle size distribution was used to characterize the base powders. The investigation results are presented in the form of histograms, showing the particle size distributions. The particle sizes of Al_2_O_3_, Ti, and Ni powders were determined using a Laser Scattering Particle Size Distribution Analyzer LA-950 HORIBA (HORIBA Instruments Inc., Irvine, CA, USA). Water was used as a dispersing medium in the trial to examine Al_2_O_3_ powder, while Ti and Ni powders were examined using isopropanol.

X-ray diffraction (XRD) phase analysis was performed to determine the phase composition of the raw powders. Moreover, the phase composition studies were carried out for the composites in the raw state and after sintering. This allowed estimation of the intermetallic phases in the investigated series of composites, especially in the sintered composites. The phase composition was determined on a Rigaku MiniFlex II X-ray diffractometer (Rigaku Corporation, Tokyo, Japan). Cu Kα radiation with wavelength λ = 1.54178 Å was used. The experiments were performed at 30 kV and 15 mA current, in the angular range of 20°−100°, in 0.02° steps, with a counting time of 2 s. The identification of the phases of the powders was carried out using JADE (MaterialsData, CA, USA) (a program for data visualization, preprocessing, and diffractogram identification) and the ICDD PDF−2/4, ICSD/NIST database (MaterialsData, CA, USA). A helium pycnometer was used to determine the actual density of the powders. Density measurement was determined based on the standard ASTM D3766. The density determined by this method is the ratio of the mass of the solid material to the sum of the volumes of solid material and closed pores within the material. The test was carried out at 700 cycles and ten rinses in the presence of helium using a Micrometrics Accu Puc II helium pycnometer (Micrometrics, Norcross, GA, USA).

The second part of the investigation was an investigation on slurries and composites in raw and sintered states.

### 2.2. Slurries

To produce slurries, the powders were weighed in appropriate proportions and mixed with the liquefiers and solvent. To prepare the slurries, citric acid (CA), (Sigma-Aldrich, St. Louis, MO, USA), and diammonium citrate (DAC), (Sigma-Aldrich, St. Louis, MO, USA), were used as liquefiers. Demineralized water was used as the solvent. The order of adding the individual components in the slurries was as follows: the first step was to dissolve the appropriate amount of citric acid and diammonium citrate liquefiers in water. The amount of added liquefiers was selected experimentally, and for all investigated composites was 0.3 wt.% DAC and 0.1 wt.% CA, relative to the amount of solid phase in the slurries. Then alumina and metallic powders were added to the prepared suspensions. The suspensions were then mixed and degassed in a THINKY ARE−250 homogenizer (Thinky Corporation, Tokyo, Japan). Mixing was carried out at 1000 rpm for 8 min while degassing at 2000 rpm for 2 min. After mixing, the slurries were cast into glass cylinders and subjected to macroscopic observations to determine their stability in time. In the present study, two slurries were prepared from the Al_2_O_3_/Ti/Ni system with a 15 vol.% metallic phase content (including a 1:1 ratio of metallic components) and different solid-phase contents: 35 vol.% (Series I) and 50 vol.% (Series II). The characteristics of the composite series involved in the study are shown in Table 1.

Analysis of the sedimentation tendency of slurries was carried out. Detailed observations were carried out for one hour from when the suspension was placed in a glass vessel. During the formation of composites by slip casting, it is essential that the slurry is stable from the time of its preparation until it is cast into the gypsum mold, which is about 40 min after homogenization. Glass vessels with suspensions were photographed at specific time intervals during observation.

Afterward, the slurries’ dynamic viscosity and shear stress as a function of shear rate were measured in two stages. In the first stage, the measurements were carried out at an increasing shear rate from 0.1 s^−1^ to 100 s^−1^. In the second stage (immediately after the first stage), the measurements were carried out at a decreasing shear rate from 100 s^−1^ to 0.1 s^−1^. Rheological examinations of the slurries were taken out using a Kinexus Pro rheometer (Malvern Panalytical, Malvern, UK) in a plate-to-plate system. The measuring gap diameter was 0.5 mm in each case.

The Al_2_O_3_/Ti/Ni composites reinforced by intermetallic phase composites were obtained by the slip casting method. This technique consists of pouring the prepared slurries into a porous mold that, under the capillary force’s action (vacuum) in the pores, extracts water from the suspension. The slurry components and the liquefaction process were optimized to obtain a slurry free from air bubbles, with adequate flowability and low shrinkage with no tendency to sediment larger grains during storage. The stable slurry can achieve composites in the raw state with favorable properties, which do not change after the sintering process.

### 2.3. Composites

In their raw state, the molded composite samples were dried in an oven at 40 °C for 48 h. The samples were then ground on SiC paper with the following gradations: 120, 400, 600, 800, 1200 to obtain a shape with flat parallel surfaces (elimination of the shrinkage cavity). Then the sample was sintered at 1400 °C in an Ar atmosphere. During the sintering, the heating and cooling rates were 2 °C min^−1^. The dwell time was 2 h.

The mechanical properties of the sintered composites were investigated. In addition, a monotonic compression test was carried out on a 5000 kN hydraulic pulsator. Results of the tests are presented in plots, which show the compression strength of the specimen (MPa) as a function of the displacement of the compression plate.

Microstructural investigations were performed on the fractures after monotonic compression tests. The microstructure was observed using a scanning electron microscope (SEM) equipped with an SE (secondary electron detector) and BSE (backscattered electron detector). The tests were done using a JSM−6610 scanning electron microscope (JEOL, Tokyo, Japan). The tests were performed at an acceleration voltage of 15 kV. In addition, surface microanalyses of chemical composition were performed using an energy dispersive X-ray spectrometer (EDX). The EDX (Oxford, UK) results were generated in maps of elemental distribution on the studied sample surface. To obtain the SEM images, metallization (sputtering) of the samples was necessary. The samples were coated with a thin layer of carbon using QUORUM Q150T ES Sputter Coater (Headquarters, Laughton, East Sussex, UK)

## 3. Results and discussion

### 3.1. Base Powders

At the first step of the study, characterization of the base powders was performed. Table 2 shows the parameters of the base powders given by the manufacturers and obtained by laser analysis of the particle size distribution. The particle size distributions of the base powders: Al_2_O_3_, titanium, and nickel, as obtained by laser particle size distribution analysis, are unimodal for all powders. The average particle size of alumina was 0.218 ± 0.05 µm (Figure 1A), which does not match with the manufacturer’s data (100 ± 25 nm). A similar result was revealed for nickel powder where the average particle size was 206.78 ± 67.71 µm (Figure 1B), which also does not match the data provided by the manufacturer (range 149 µm to 297 µm). The difference between the average size measured by the dynamic light scattering method and the manufacturer’s data is probably due to the formation of agglomerates by the nanometric and micrometric powder particles. For the titanium powder, the average particle size was 271.39 ± 115.10 µm (Figure 1C), within the range of particle sizes measured by the manufacturer (149–250 µm). The density of the alumina powder determined by the pycnometric method was 3.896 g/cm^3^, which is lower than the manufacturer’s reported density of 3.98 g/cm^3^. For the titanium and nickel powders, the measured densities were 4.431 g/cm^3^ and 8.733 g/cm^3^, respectively, which are also lower than the densities specified by the manufacturer (4.506 g/cm^3^ and 8.9 g/cm^3^, respectively). The mismatch in the results is due to a measurement error associated with insufficient desorption of the powder before measurement. This discrepancy is typical for nano- and micro-powders that exhibit high surface development. When full desorption is not possible, the volume may be overestimated or underestimated. Thus, the measured density is lower when the powder volume is overestimated and higher when the volume is underestimated. In addition, the results can be incorrect due to contaminants that may be deposited on the surface of the test powder during preparation for measurement.

Figure 2 shows the morphology of the starting ceramic and metallic powders visualized by the scanning electron microscope. SEM micrograph analysis showed significant diversification in the morphology of the powders applied in the study. It was observed that both Al_2_O_3_ and nickel powders (Figure 2A,B) featured a quite regular morphology with rounded edges and shapes mostly close to spherical. It has also been noticed that the surface of the nickel particles is rough with a lot of cavities. Furthermore, it was observed that the Al_2_O_3_ powders are characterized by the tendency to form agglomerates. The direct observation of the metal powders allowed us to find that titanium powder shows a highly irregular shape (Figure 2C).

Figure 3 shows the results of phase analysis of base powders. Phase analysis results of all base powders revealed the absence of peaks coming from other phases, which confirms the high purity of alumina, nickel, and titanium powders declared by the manufacturer. The obtained results were analyzed using PDF data sheets: alumina-04–013−1687, nickel-04−016−4761, and titanium 04−004−4447.

### 3.2. Slurries

The next step was to research with slurries containing 35 vol.% (Series I) and 50 vol.% (Series II) solid phase. Both series contained 15 vol.% metallic phase. The experiment consisted of detailed observations of the glass vessels with suspension for one hour from when the slurries were placed in the vessels. The results are shown in Figure 4. Observation of the slurries did not reveal any changes in appearance during the experiment. Thus, all slurries subjected to macroscopic observations showed stability over time.

The rheological properties of the slurries used in the study were investigated. From the hysteresis loops visible in the graph shown in Figure 5A, it can be concluded that both suspensions are thixotropic fluids. Moreover, the suspensions are characterized by small values of the flow limit below 2 Pa. The flow limit is the shear stress value for the shear rates near 0 s^−1^. Based on the viscosity curves for high (Figure 5A) and low (Figure 5B) shear rates, it can be concluded that both analyzed suspensions are shear-thinning fluids. Their viscosity decreases with increasing shear rate as the proportion of solids in the suspension increases, their viscosity increases. The viscosity values at a 10 s^−1^ shear rate are 0.041 and 0.429 Pa∙s for suspensions with 35% and 50% solid phase content, respectively. Small viscosity values indicate that the slurry forming process was performed correctly.

In the present study, gypsum molds were used for composites molding because of their low cost, excellent shape reproduction, and surface smoothness. Figure 6 shows the macro and microstructure of the gypsum mold. A cylindrical gypsum mold of appropriate porosity was produced. It was allowed to control the drainage of liquid medium from the prepared slurries and thicken suspension. In the present study, class II white gypsum -Student from Zhermack was used. The mixing ratio used to make the gypsum was: 50 mL of water/100 g gypsum. The obtained gypsum mold was characterized by a compressive strength >9 MPa and linear expansion after two hours <0.30%. The slurries were then cast into the gypsum molds.

### 3.3. Composites

In the next step of research, composites in the raw state were characterized. Figure 7 shows typical samples obtained by the slip casting method in the raw state. Macroscopic observations of the composites in the raw state revealed composites without visible defects in the form of cracks, microcracks, or delaminations on their surfaces.

Figure 8 shows the phase analysis results from the surfaces of both composite series, with solid phase contents of 35 vol.% and 50 vol.%, respectively. Analysis of the obtained results was carried out using PDF sheets: aluminum oxide-04−015−8995, titanium-01−088−2321, and nickel-04−006−6387. Analysis of the diffractograms revealed only three phases in all composites in the raw state-aluminum oxide, titanium, and nickel. In addition, no peaks indicated the presence of other new phases in the composites.

The next study was the characterization of the sintered composites. The XRD results of the investigated composites revealed a TiNi intermetallic phase in the composite containing 35 vol.% solid phase (Figure 9A). In the case of the composite containing 50 vol.% solid phase (Figure 9B), two types of intermetallic phases were revealed: TiNi and Ni_3_Ti. The sintering temperature (1400 °C) causes the melting of the vast majority of Ti-Ni phases (including intermetallic) [30]. For this reason, the only time these compounds can form is during the cooling stage at temperatures starting from 1380 °C, 1310 °C, and 984 °C for TiNi_3_, TiNi, and Ti_2_Ni, respectively [30]. Therefore, in the 50/50 Ti–Ni system, the TiNi phase should be intermetallic with the highest participation, corresponding to the obtained results. At the same time, an inevitable local heterogeneity in the concentration of Ti and Ni in the metallic phase can promote the formation of other phases (e.g., nickel-rich TiNi_3_).

Figure 10 shows the results of the monotonic compression test for the two series of composites. The character of the plots was similar for both composites: in the first stage of loading, a slight slope is noticeable, then the plots become more linear, and in the final stage, a decrease in load occurs. This force decrease was caused by the initiation of a crack in the test specimen. The peak values of the compressive load were determined using the maximum compressive stress obtained during the test. The results were used to calculate the compression strength (K) of the composites. The results of the calculations are shown in Table 3, where the highest compression strength value (K = 4.37 MPa) was obtained for the specimen containing 50 vol.% solids. Lower compression strength was revealed for the composite containing 35 vol.% solid phase (K = 3.78 MPa). The decrease in compression strength by about 13% and slightly better ductility can be explained by the lower proportion of ceramic in favor of the metallic phase. The Authors previously published research results of monotonic compression testing of an Al_2_O_3_/Ni composite [36] indicated that the maximum compressive strength was 42.45 MPa which is about ten times higher than the strength of the Al_2_O_3_/Ti/Ni (3.78 MPa for Series I and 4.37 for Series II) presented in this paper. This significant difference can be attributed to the presence of sintered brittle intermetallic phases and the fact that the bonds between the alumina matrix and metallic particles are the main fracture mechanism confirmed by fractography observation results.

The SEM images (in backscattered electron (BSE) mode) in Figure 11A,B show the characteristic areas of sintered composites with different solid phase content. Red and yellow marked areas correspond to magnified areas. Microstructural studies were performed on the fractures obtained after monotonic compression tests. In BSE mode, the alumina phase is shown as dark grey areas, while the metallic phases are shown as light grey areas. Based on the SEM observation, it can be concluded that the particles of the metallic phase are dispersed throughout the volume of the composites samples. An irregular shape characterizes the metallic phases. Their distribution is fairly uniform within the samples. The microstructural analysis does not reveal areas characterized by enrichment or lack of metallic phases. Composites are characterized by a relatively uniform shape and size of the metallic phase particles, regardless of the content of the metallic phase. Fractography analysis of the samples revealed that the bonds between the alumina matrix and the metallic phase are the weakest points of the composite. Observation of cracks between them leads to the conclusion that debonding of the alumina matrix and metallic particles is the main fracture mechanism of the investigated composites. Fracture observation supports the conclusion that the intergranular fracture mechanism characterizes the composites. The cracks in the alumina reveal well-developed grain boundaries. Similar fracture patterns are characterized for both investigated series.

EDS analysis results in the form of the element distribution maps shown in Figure 12A,A’,B,B’ confirmed the chemical composition of the light and dark areas in the SEM images as metals, solid solution, and Al_2_O_3_, respectively. The distribution of the elements on the fracture surface revealed a solid solution of nickel and titanium in both studied samples. This was confirmed by the EDS analysis of the characteristic areas visible in Figure 12. The results of the chemical composition analysis are presented in Table 4. Images A and B correspond to the areas with separated particles of Ni and Ti while A’ and B’ images reflect areas composed of both elements, titanium and nickel, which are in the solid solution state. EDS measurements did not allow us to identify whether the analysed area is an intermetallic or solid state. Therefore, previously presented XRD measurements were applied (Figure 9). Uniformly distributed areas of pure titanium and nickel were also observed.

## 4. Conclusions

The results provided new fundamental knowledge about ceramic–metal composite materials of Al_2_O_3_/Ti/Ni system formed by the slip casting method and reinforced with intermetallic phases.

Rheological studies showed that all funnel masses are shear-thinning suspensions. The lack of sedimentation tendency of the obtained suspensions showed that they were properly composed and capable of producing composite samples from the Al_2_O_3_/Ti/Ni system.

The slip casting method allows the production of composites containing aluminum oxide, titanium, nickel, and TiNi and Ni_3_Ti intermetallic phases. The particles of the metallic phase are dispersed throughout the volume of the composite samples. An irregular shape characterizes the metallic phases.

Macroscopic observations of the sintered composites after the monotonic compression tests revealed intergranular fractures. The results of the compression test study revealed the highest compression strength value (K = 4.37 MPa) for the samples containing 50 vol.% solids. A lower compression strength was found for the composite containing 35 vol.% solid phase (K = 3.78 MPa).

The developed methodology for obtaining ceramic–metal composites reinforced with intermetallic phases provides a starting point for application-related work. The results obtained are of high scientific value and application potential. This research represents the development of a technological basis for the production of new innovative Al_2_O_3_/Ti/Ni composites reinforced with intermetallic phases. Work on this subject is in progress and new results will be published in succeeding articles.

## Figures and Tables

**Figure 1 materials-14-03510-f001:**
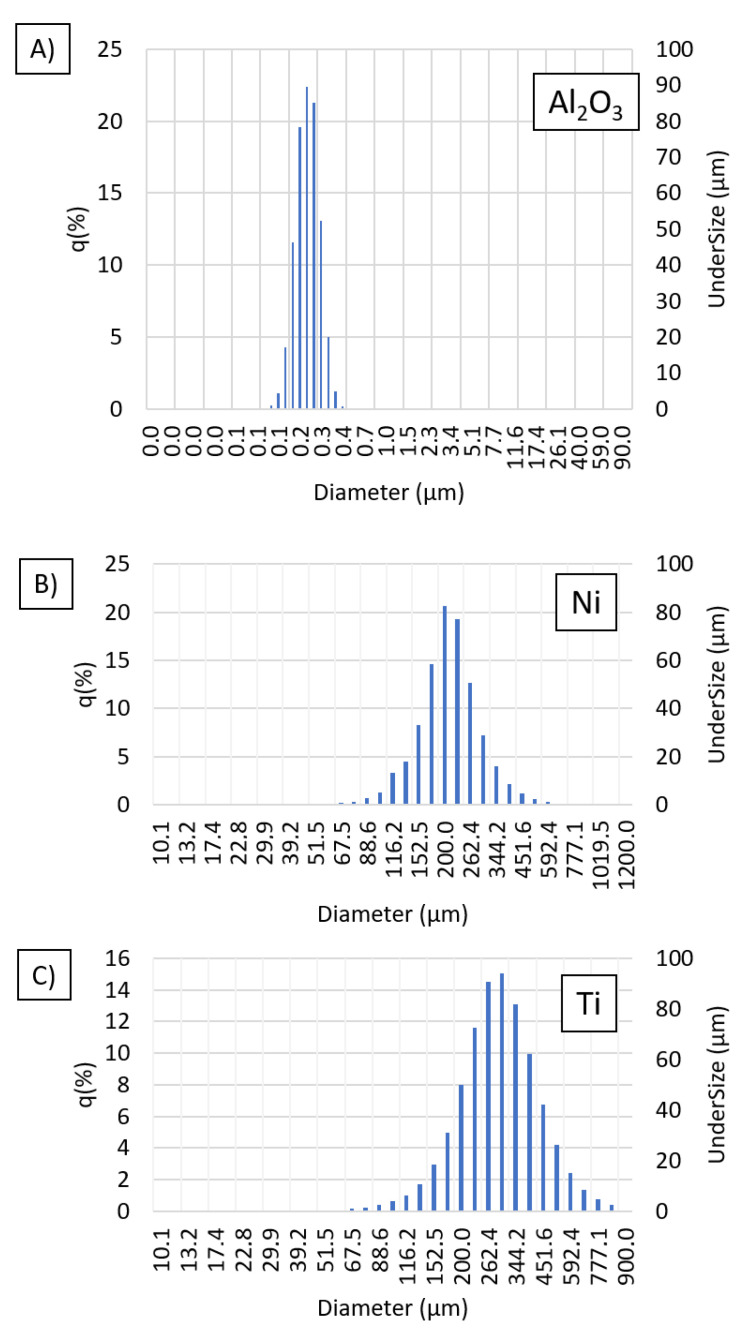
Histogram of particle size distribution results for (**A**) alumina powder, (**B**) nickel powder, and (**C**) titanium powder.

**Figure 2 materials-14-03510-f002:**
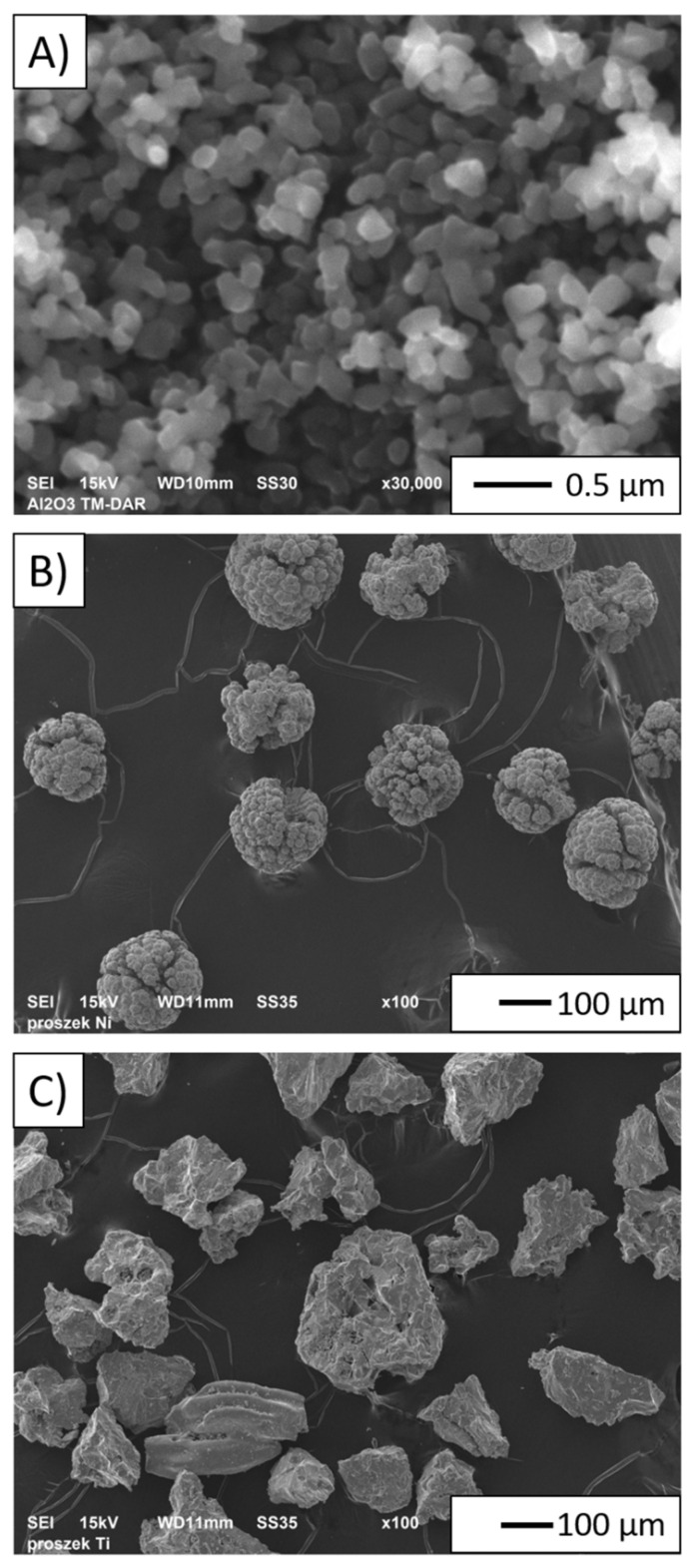
Microstructure of the powders: (**A**) aluminium oxide; (**B**) nickel; (**C**) titanium.

**Figure 3 materials-14-03510-f003:**
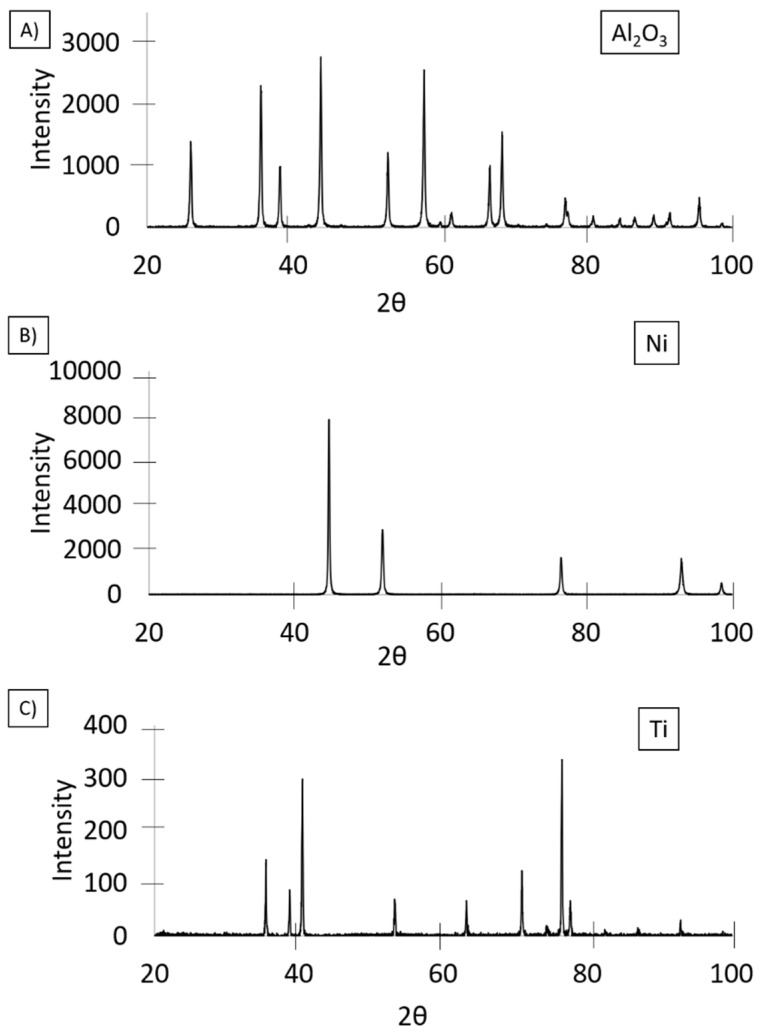
X-ray diffractograms of (**A**) alumina powder, (**B**) nickel powder, and (**C**) titanium powder.

**Figure 4 materials-14-03510-f004:**
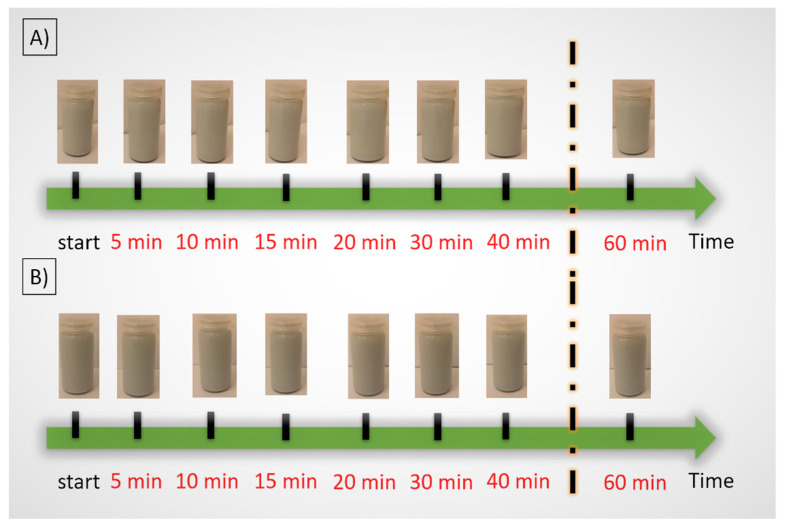
Macroscopic observation of the slurries, (**A**) containing 35 vol.% (Series I) and (**B**) 50 vol.% (Series II) solid phase.

**Figure 5 materials-14-03510-f005:**
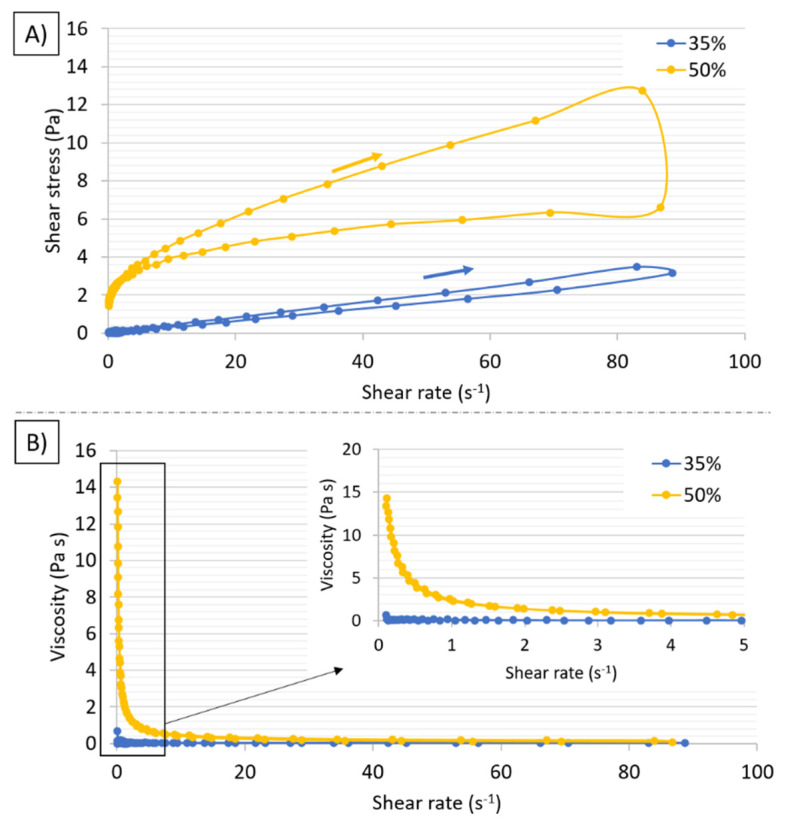
Viscosity curves of the slurries for (**A**) viscosity curves, and (**B**) flow curves.

**Figure 6 materials-14-03510-f006:**
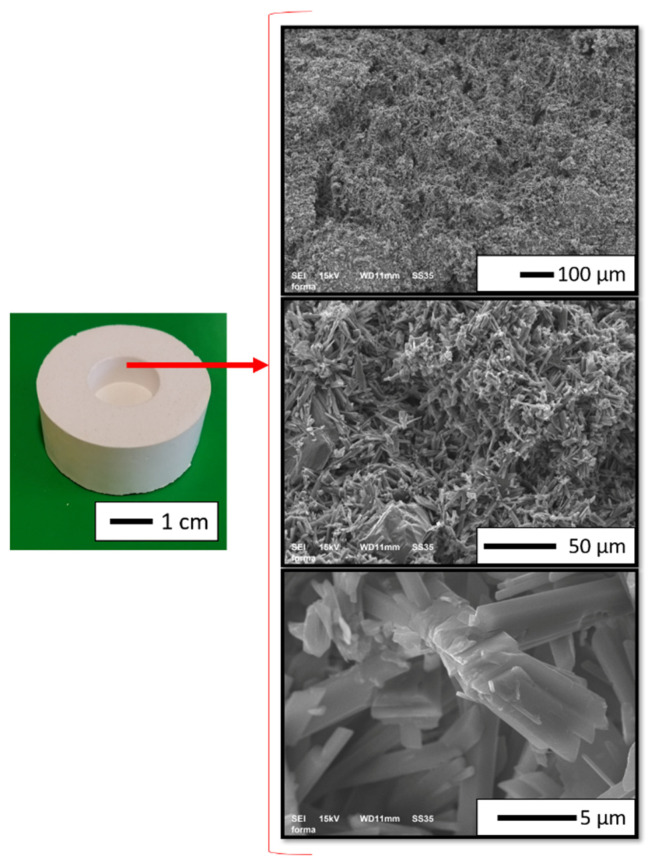
Macro and microstructure of the gypsum mold.

**Figure 7 materials-14-03510-f007:**
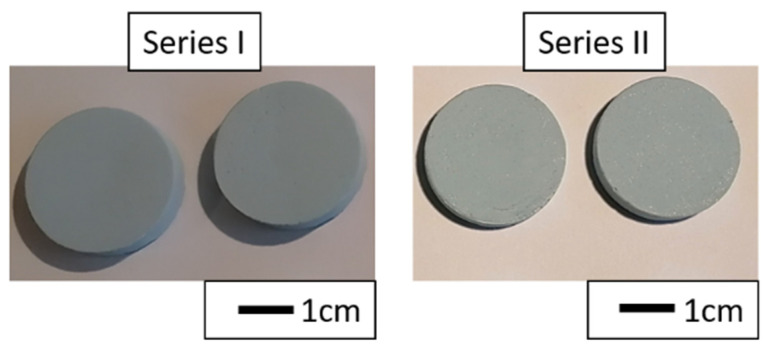
Typical samples obtained by the slip casting method in the raw state.

**Figure 8 materials-14-03510-f008:**
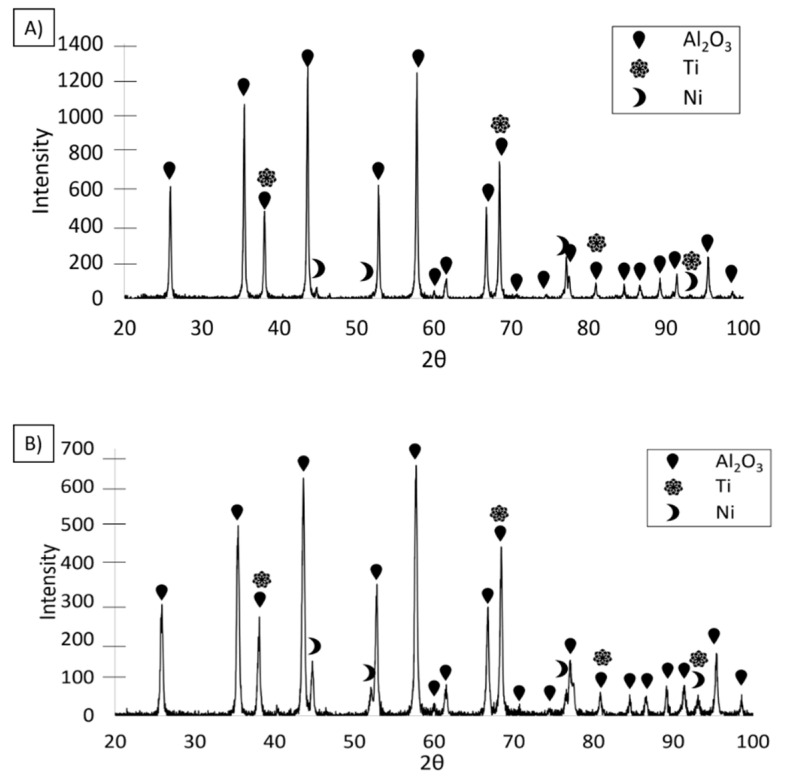
Diffractograms of the composites in the raw state with solid contents of (**A**) 35 vol.%, and (**B**) 50 vol.%.

**Figure 9 materials-14-03510-f009:**
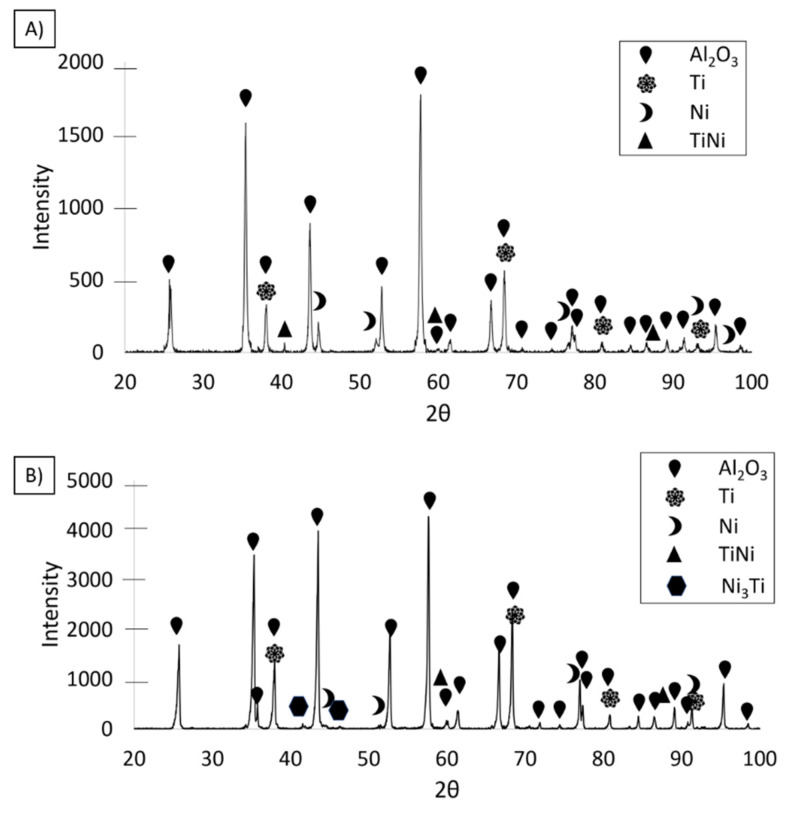
The phase composition of the sintered composite from (**A**) Series I, and (**B**) Series II.

**Figure 10 materials-14-03510-f010:**
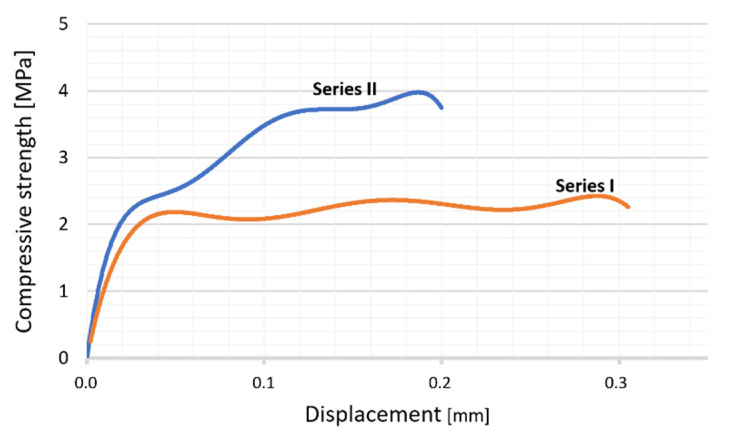
Stress–displacement graph of the composites: orange line—Series containing 35 vol.% solid phase (Series I); blue line—Series containing 50 vol.% solid phase (Series II).

**Figure 11 materials-14-03510-f011:**
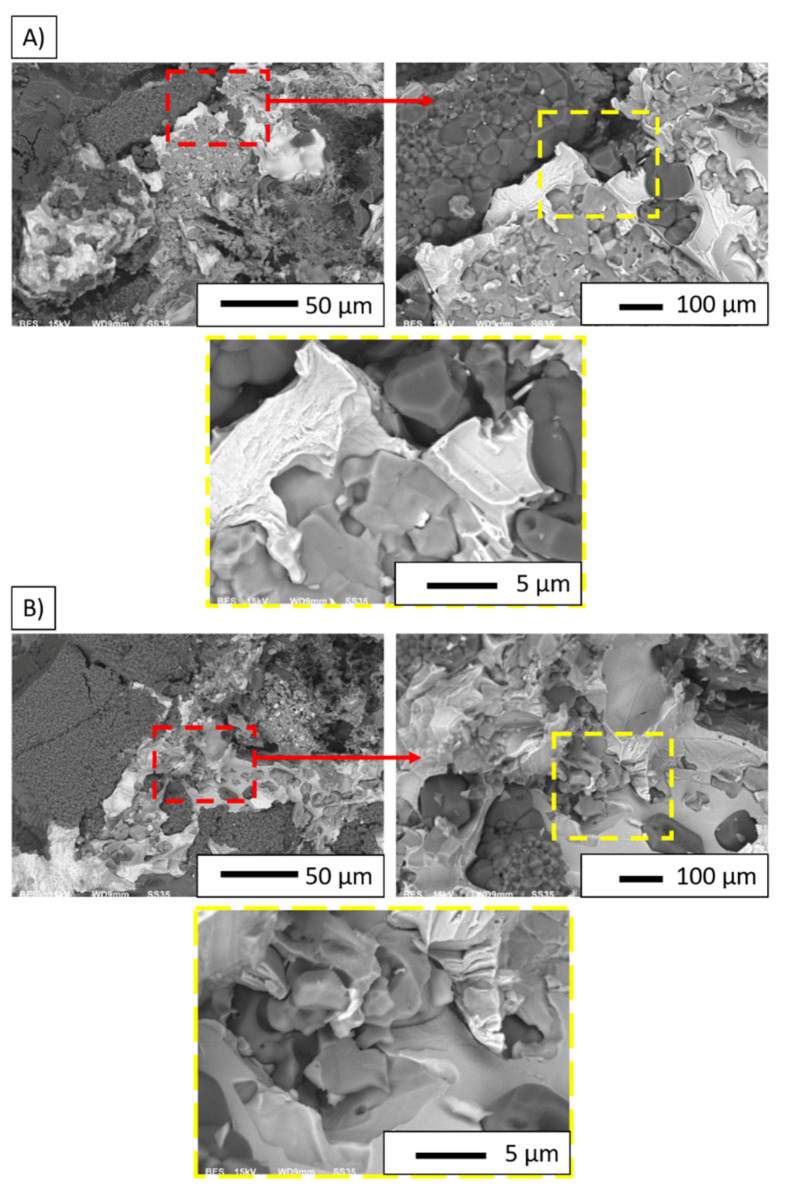
SEM images of fractographic observation of samples: (**A**) Series I; (**B**) Series II.

**Figure 12 materials-14-03510-f012:**
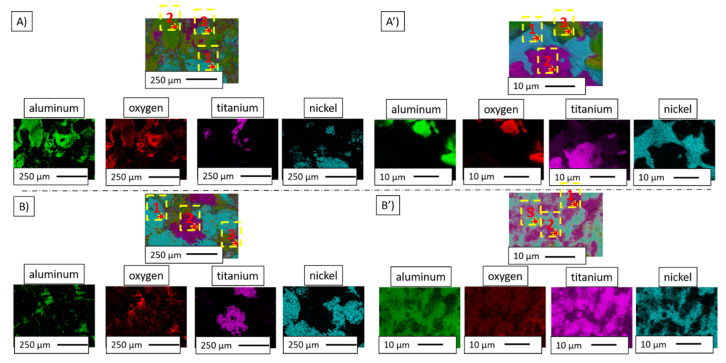
Chemical element distribution maps for samples: (**A**,**A’**) Series I; (**B**,**B’**) Series II.

**Table 1 materials-14-03510-t001:** Characteristics of the composite series.

Series No.	Chemical Composition	Metallic Phase Content [vol.%]	Solid Content [vol.%]	Titanium/Nickel Ratio [%]
1.	Al_2_O_3_/Ti/Ni	15	35	50/50
2.			50	

**Table 2 materials-14-03510-t002:** Parameters of raw powders.

Parameter	α-Al_2_O_3_ (Taimei Chemicals)	Ti (Alfa Aesar)	Ni (Alfa Aesar)
Average particle size [µm]	0.100 ± 0.025	149–250	149–297
Average particle size from PSD analysis (Particle Size Distribution) [µm]	0.218 ± 0.05	271.39 ± 115.10	206.78 ± 67.71
Density [g/cm^3^]	3.980	4.506	8.900
Density was measured with a pycnometer [g/cm^3^]	3.896	4.431	8.733
Purity [%]	99.9	99.5	99.7

**Table 3 materials-14-03510-t003:** Monotonic compression test results.

Series No.	Solid Content [Vol. %]	Maximum Compression Load F [N]	Compression Strength K [MPa]
1.	35	378	3.78
2.	50	802	4.37

**Table 4 materials-14-03510-t004:** Chemical composition of characteristic areas of the composites. Measurement areas are shown in Figure 12.

Series	Image	Point	Chemical Composition							
			O		Al		Ti		Ni	
			%wt.	%at.	%wt.	%at.	%wt.	%at.	%wt.	%at.
I	A	1	2.10	7.24	0.91	1.85	---	---	96.99	90.91
		2	37.62	50.93	59.95	48.12	0.66	0.30	1.77	0.65
		3	1.66	5.69	2.05	4.18	---	---	96.29	90.13
	A’	1	1.72	5.74	1.87	3.71	13.60	15.17	82.81	75.37
		2	18.20	40.00	0.46	0.60	79.16	58.10	2.18	1.30
		3	46.73	60.60	49.00	37.68	2.63	1.14	1.64	0.58
II	B	1	1.26	4.44	0.77	1.61	---	---	97.97	93.95
		2	33.54	59.28	3.64	3.82	61.07	36.06	1.75	0.84
		3	35.77	52.61	45.27	39.48	3.44	1.69	15.51	6.22
	B’	1	20.32	43.29	2.99	3.77	64.11	45.63	12.59	7.31
		2	---	---	13.56	24.20	26.43	26.57	60.02	49.23
		3	21.58	45.83	3.20	3.94	60.07	41.66	15.14	8.57

## Data Availability

Data sharing not applicable.

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
