# Peer review of "Manufacturing of Al2O3/Ni/Ti Composites Enhanced by Intermetallic Phases"

_materials, 2021, doi:10.3390/ma14133510_

Round 1

Reviewer 1 Report

The paper presents a series of information manufacturing of Al2O3/Ni/Ti composites enhanced by intermetallic phases

From the analysis of the information presented in the article, I found the following:

- The paper presents a series of results that may be of interest to the scientific community:

- The introductory part should be improved: references to bibliography referring to many bibliographic sources at the same time should be avoided ([4-8], [9-13] etc.). Also, I did not identify in the paper the reference to the bibliographic source 30;

- Research methodology is not properly explained. Thus, the motivation to choose a certain composition of materials must be explained more clearly;

- Additional information on applied sintering technology needs to be added;

- The scientific part of the paper needs to be substantially improved, because in this form the paper looks more like a technical report;

- Experimental data are inadequately analyzed and more emphasis should be placed on establishing the causes that led to their obtaining;

- The discussion part needs to be substantially improved as the novelty of the results obtained in the research compared to other current results in the field is not highlighted in this section;

- The conclusions should be more concrete and include a range of information on the areas of use of these composite materials, as well as future research directions.

Author Response

Ref: Manuscript ID: materials-1256173

Title: “Manufacturing of Al2O3/Ni/Ti composites enhanced by intermetallic phases.”

Journal: Materials

On behalf of co-authors and my own, I would like to thank the reviewer for comments that are useful for the improvement of the article. I would like to explain the changes made in the revised version of the manuscript following the reviewer's suggestions.

Author's Reply to the Review Report (Reviewer 1):

“The paper presents a series of information manufacturing of Al2O3/Ni/Ti composites enhanced by intermetallic phases

From the analysis of the information presented in the article, I found the following:

- The paper presents a series of results that may be of interest to the scientific community:

- The introductory part should be improved: references to bibliography referring to many bibliographic sources at the same time should be avoided ([4-8], [9-13] etc.). Also, I did not identify in the paper the reference to the bibliographic source 30;”

Answer:

The authors of the manuscript thank the reviewer for taking the time to review. The introduction is improved. References are separated and a few new are added. Identification of 30 bibliographic source has been added in the text, as well. All changes made in the revised version of the manuscript are marked with red colour in the text.

“- Research methodology is not properly explained. Thus, the motivation to choose a certain composition of materials must be explained more clearly;”

Answer:

Thank you for the remark.

The motivation to choose a certain composition of materials was designed according to our experience with the previous investigation of Al2O3, Al2O3/Ni, and Al2O3/Ti. In the experiment, the suspensions used to produce the specimens were characterized by a solid phase content of 35 vol% and 50 vol%. The above solids contents were selected to test how the solids content of the slip would affect the obtained composites. The choice of the Al2O3 TM-DAR powder was mainly managed by their very high purity and submicrometric particle size. In addition, these powders are often used by scientists, thus providing for the following comparisons to be made with international study results. Moreover, when selecting alumina powder as the ceramic matrix, the authors were also guided by the fact that Al2O3 characterize by high hardness, satisfactory strength properties, high chemical, and thermal shock resistance, and low cost in comparison to other ceramic materials. In addition, Al2O3 can be considered as one of the most widespread ceramic oxide materials in advanced ceramics manufacturing. The use of ceramic powder with submicron micrometer particle size enables the prepared slurry to liquefy easily. This is very important when preparing samples by slip casting. On the other hand, the choice of Ni and Ti powders was dictated by the fact that they easily form intermetallic phases. Moreover, the chosen metal powders are characterized by good thermal and electrical conductivity, ductility, and corrosion resistance. Obtaining intermetallic phases in composites formed by slip casting was the main goal of the research. The selection of the sintering temperature (1400°C) was also not accidental. The sintering temperature (1400°C) causes the melting of the vast majority of Ti-Ni phases (including intermetallic). For this reason, the only possibility of these compounds to form is during the cooling stage in the temperature range, starting from 1380°C, 1310°C, and 984°C for TiNi3, TiNi, and Ti2Ni, respectively. Therefore, in the 50/50 Ti-Ni system, the TiNi phase should be intermetallic with the highest participation, corresponding to the obtained results. At the same time, an inevitable local heterogeneity in the concentration of Ti and Ni in the metallic phase can promote the formation of other phases (e.g., nickel-rich TiNi3).

Information about what led the authors during the selection of powders for production composites has been added to the manuscript.

“- Additional information on applied sintering technology needs to be added;”

Answer:

Thank you for your comment. In the test added information about the dwell time during sintering. In addition, the manuscript contains information about the heating and cooling time during the sintering process, as well as the sintering temperature and the type of atmosphere in which the sintering was carried out. All changes made in the revised version of the manuscript are marked with red colour in the text.

“- The scientific part of the paper needs to be substantially improved, because in this form the paper looks more like a technical report;”

Answer:

The authors agree with the reviewer that the scientific part of the paper needs to be substantially improved, because in this form the paper looks more like a technical report. Therefore, the authors changed the content of the manuscript in accordance with the reviewer's remark. All changes made in the revised version of the manuscript are marked with red colour in the text.

“- Experimental data are inadequately analyzed and more emphasis should be placed on establishing the causes that led to their obtaining;”

Answer:

Thank you very much for this suggestion. The authors agree with the reviewer that experimental data are inadequately analyzed and more emphasis should be placed on establishing the causes that led to their obtaining. The authors of the manuscript made every effort to change the manuscript according to the suggestion by the reviewer. All changes made in the revised version of the manuscript are marked with red colour in the text.

“- The discussion part needs to be substantially improved as the novelty of the results obtained in the research compared to other current results in the field is not highlighted in this section;”

Answer:

The Authors agree that the novelty of the results is not compared to other current results. Authors decided to add comparison of the monotonic compression test results with the previously published result for the Al2O3/Ni composite (J. Zygmuntowicz et al. Composites Part B 156 (2019) 113–120). The maximum compressive strength was 42.45 MPa what is about ten times more value than the strength of the Al2O3/Ti/Ni (3.78 MPa for Series I and 4.37 for Series II) revealed in the paper. This significant difference can be due to the presence of sintered brittle intermetallic phases and the fact that the bonds between the alumina matrix and metallic particles are the main fracture mechanism confirmed by fractography observation results.

“- The conclusions should be more concrete and include a range of information on the areas of use of these composite materials, as well as future research directions.”

Answer:

The authors agree with the reviewer that the conclusions should be more concrete and include a range of information on the areas of use of these composite materials, as well as future research directions. Consequently, the authors changed the conclusions in accordance with the reviewer's comments. All changes made in the revised version of the manuscript are marked with red colour in the text.

All changes made in the revised version of the manuscript are described above and marked with red color in the text. I again appreciate the help from the reviewers in improving the manuscript and hope the paper will be accepted for publication.

Yours sincerely,

Justyna Zygmuntowicz

Warsaw University of Technology

Faculty of Materials Science and Engineering

141 Woloska Str.

02-507 Warsaw

Poland

Reviewer 2 Report

In this manuscript, the authors reported on the Al2O3/Ti/Ni ceramic-metal composites prepared by using the slip casting method. The base powders, slurries, and sintered composites were analyzed in detail. They found that the slip casting method allowed obtaining composites enhanced by intermetallic phases. I think it can be accepted after a few major issues are addressed.

  1. The whole manuscript was written like an experiment report, not a journal article. I suggest to divide the “Materials and Methods” and “Results and discussion” into three parts, such as base powders, slurries and composites, and characterizations.
  2. Line 251-253, Why is TiNi only in the composite containing 35 vol.% solid phase without other intermetallic phase? Slurry is not stable over time or dispersion is not uniform?
  3. Line 308-313, the mapping of the elements on the fracture surface strongly depends on the flatness of the fracture surface. Therefore, I can not accept the description here. Please provide the elements composition in atomic percentage in Figure 12 (right pictures), Al-, O-, Ti- and Ni-rich areas, which may be consistent with the XRD results in Figure 9.
  4. The authors described “The conducted research revealed that the slip casting method allows obtaining composites enhanced by intermetallic phases…”. I think you need a pure Al2O3 sample as a reference.
  5. For the Figure 3, add the standard XRD cards for ensuring no impurities. How about vol.% for DAC and CA in Line 100? What is the “solid phase” when the authors described the content? What is the dwell time for sintering composites at 1400 °C?
  6. For the Figure 5, the caption is not the same with the figure content. Line 230, “50 ml of water / 100 gypsum”, what is the unit? What is the size of composites in raw and sintered states? Did the authors consider the shrinkage of samples? Figure 8b, what is peak at 2θ ≈ 40? Add reference at Line 258.
  7. Some repeated descriptions need to be avoided (Line 140-141 and Line 266-267, Line 146 and 304, etc.). Please check throughout the whole paper.

Author Response

Ref: Manuscript ID: materials-1256173

Title: “Manufacturing of Al2O3/Ni/Ti composites enhanced by intermetallic phases.”

Journal: Materials

On behalf of co-authors and my own, I would like to thank the reviewer for comments that are useful for the improvement of the article. I would like to explain the changes made in the revised version of the manuscript following the reviewer's suggestions.

Author's Reply to the Review Report (Reviewer 2):

“In this manuscript, the authors reported on the Al2O3/Ti/Ni ceramic-metal composites prepared by using the slip casting method. The base powders, slurries, and sintered composites were analyzed in detail. They found that the slip casting method allowed obtaining composites enhanced by intermetallic phases. I think it can be accepted after a few major issues are addressed.

  1. The whole manuscript was written like an experiment report, not a journal article. I suggest to divide the “Materials and Methods” and “Results and discussion” into three parts, such as base powders, slurries and composites, and characterizations.”

Answer:

The authors agree with the Reviewer that the separation of the “Materials and Methods” and “Results and discussion” on three sections will significantly increase the quality of the manuscript. The authors have made a change. All changes made in the revised version of the manuscript are marked with red colour in the text.

  1. “Line 251-253, Why is TiNi only in the composite containing 35 vol.% solid phase without other intermetallic phase? Slurry is not stable over time or dispersion is not uniform?”

Answer:

The XRD results of the investigated composites revealed TiNi intermetallic phase in the composite containing 35 vol.% solid phases only. In the case of the composite containing 50 vol.% solid phase, two types of intermetallic phases were revealed: TiNi and Ni3Ti. The sintering temperature (1400°C) causes the melting of the vast majority of Ti-Ni phases (including intermetallic). For this reason, the only possibility of these compounds to form is during the cooling stage in the temperature range, starting from 1380°C, 1310°C, and 984°C for TiNi3, TiNi, and Ti2Ni, respectively. Therefore, in the 50/50 Ti-Ni system, the TiNi phase should be intermetallic with the highest participation, corresponding to the obtained results. At the same time, an inevitable local heterogeneity in the concentration of Ti and Ni in the metallic phase can promote the formation of other phases (e.g., nickel-rich TiNi3).

Moreover, answering the question "The slurry is not stable over time, or dispersion is not uniform?”, the sedimentation tendency of slurries was carried out. The experiment consisted of detailed observations of the glass vessels with suspension for one hour from when the slurries were placed in the vessels. Observation of the slurries did not reveal any changes in appearance during the experiment. Thus, all slurries subjected to macroscopic observations showed stability over time.

  1. “Line 308-313, the mapping of the elements on the fracture surface strongly depends on the flatness of the fracture surface. Therefore, I can not accept the description here. Please provide the elements composition in atomic percentage in Figure 12 (right pictures), Al-, O-, Ti- and Ni-rich areas, which may be consistent with the XRD results in Figure 9.”

Answer:

The authors agree that the EDS mapping analysis should be carried out only on flat surfaces. Moreover, the authors realize that the study was conducted on the fracture, so the tested surface is not flat, must take into account that the incident radiation can bounce back from the surface and can arouse adjacent elements. The authors are fully aware that in addition, it should be remembered that on the surface of the fracture, it may be a small number of other phases which is invisible under a microscope. The authors used fractures to overview through the composites, only. To minimalize error EDS maps were carried out with the high time of measurements. According to the Reviewer's suggestion, the Authors added elements composition (%wt. and at.%) of the characteristic areas observed in Figure 12. All changes made in the revised version of the manuscript are marked with red colour in the text.

  1. “The authors described “The conducted research revealed that the slip casting method allows obtaining composites enhanced by intermetallic phases…”. I think you need a pure Al2O3 sample as a reference.”

Answer:

In this manuscript, the research aims to examine the possibility of manufacturing by slip casting and characterize the microstructure and mechanical properties of Al2O3/Ti/Ni composites reinforced with intermetallic phases. According to the authors, the production and characterization of a sample containing pure alumina will not contribute much to the article. Especially that fittings containing pure alumina (TM_DAR) have been characterized many times in the earlier works of Zygmuntowicz and her team. According to the authors, presented in the manuscript of the study will allow for the basic development of a methodology for influencing the properties of composites with a ceramic matrix by introducing a metallic phase, which in the sintering process will form an intermetallic reinforcement and be an important reference point for future work on the possibility of obtaining three-component ceramic-metal composite materials.

  1. “For the Figure 3, add the standard XRD cards for ensuring no impurities. How about vol.% for DAC and CA in Line 100? What is the “solid phase” when the authors described the content? What is the dwell time for sintering composites at 1400 °C?”

Answer:

Authors added numbers of PDF cards used in XRD research.

The dispersion was achieved by adding 0.3 wt% of diammonium hydrocitrate, DAC, and 0.1 wt% citric acids, CA referred to total solid weight, in accordance with our previous study [https://doi.org/10.1016/j.compositesb.2017.12.056; https://doi.org/10.2298/PAC1504199Z.] So far, in all previous works carried out by Zygmuntowicz's team, the values of the deflocculants were given in relation to total solid weight. Therefore, the authors believe that it is not necessary to include the volume fraction of deflocculants in the text.

In the present study, two slurries were prepared from the Al2O3/Ti/Ni system with 15 vol.% metallic phase content (including a 1:1 ratio of metallic components) and different solid-phase contents, respectively: 35 vol.% (Series I) and 50 vol.% (Series II) by volume in the prepared suspensions.

The dwell time was 2 h.

  1. “For the Figure 5, the caption is not the same with the figure content. Line 230, “50 ml of water / 100 gypsum”, what is the unit? What is the size of composites in raw and sintered states? Did the authors consider the shrinkage of samples? Figure 8b, what is peak at 2θ ≈ 40? Add reference at Line 258.”

Answer:

Thank you for your observation. The authors made a mistake in the caption of Figure 5. The authors have made a change.

Unit of the gypsum weight was gram. Authors added information in the text.

We did not determine volumetric and linear shrinkage. The authors realize that this is one of the fundamental measurements and it will be the subject of research presented in the next article.

Most likely it is noise, as no peak 2θ ≈ 40 selected by the software or manually corresponds to this angle. Diffractometric data were processed using MDI JADE 7 software (MaterialsData, CA, USA). The ICDD PDF-4+ 2020 X-ray standard database was used to interpret the results. Therefore, it was not marked on the obtained diffraction pattern.

Authors added reference to Line 258.

All changes made in the revised version of the manuscript are marked with red colour in the text.

  1. “Some repeated descriptions need to be avoided (Line 140-141 and Line 266-267, Line 146 and 304, etc.). Please check throughout the whole paper.”

Thank you very much for the important suggestion. As suggested by the reviewer repeated description of monotonic compression test and SEM/EDS research in results section was deleted.

All changes made in the revised version of the manuscript are described above and marked with red color in the text. I again appreciate the help from the reviewers in improving the manuscript and hope the paper will be accepted for publication.

Yours sincerely,

Justyna Zygmuntowicz

Warsaw University of Technology

Faculty of Materials Science and Engineering

141 Woloska Str.

02-507 Warsaw

Poland

Reviewer 3 Report

The authors reported the manufacturing of ceramic-metal composite materials Al2O3/Ti/Ni by slip casting method and studied the phase composition, microstructure, and mechanical properties of the composites. The following revisions are needed before it is well suited for publication.

The language features a large number of minor flaws, typos and mistakes, which needs to be corrected considerably to meet the high language requirement of Materials. To list just a few: line 310-311, EDS measurements do not allow to estimate of the bright areas correspond to the solid solution or intermetallic phases identified by XRD. 

The introduction is not well organized. Currently there is only one paragraph. It needs to be re-paragraphed to make it more readable.

Please include a bit more introduction about the application background of ceramic-metal composites in the very beginning of the ‘Introduction’ section. e.g., these unusual properties (taking the advantage of the properties of individual material) have made ceramic-metal composites quite promising to be used in the field of biomedical (10.1016/j.jmbbm.2017.07.024), aerospace (Composites Part B 82 (2015) 13-22) and other engineering applications.

The section ‘Materials and Methods’ can be subdivided (by adding subtitles) to make it more readable.

Fig. 1 has a rather low quality. They can hardly be seen clearly.

Add scale lines in the vertical axes in Fig. 3.

Fig. 4 has a rather low quality. The difference can hardly be seen clearly.

The vertical axes in Figs. 8-9 do not have any scale lines.

Fig. 12 has two A) and B), it is very consuming. If they are just the corresponding locally zoomed-in images, they need to be positioned properly with only one A) and B).

Normally the figure caption should be a phrase, i.e., ‘Fig. 7 The typical samples obtained by the slip casting method in the raw state’, where ‘were’ should be deleted.

Author Response

Ref: Manuscript ID: materials-1256173

Title: “Manufacturing of Al2O3/Ni/Ti composites enhanced by intermetallic phases.”

Journal: Materials

On behalf of co-authors and my own, I would like to thank the reviewer for comments that are useful for the improvement of the article. I would like to explain the changes made in the revised version of the manuscript following the reviewer's suggestions.

Author's Reply to the Review Report (Reviewer 3):

“The authors reported the manufacturing of ceramic-metal composite materials Al2O3/Ti/Ni by slip casting method and studied the phase composition, microstructure, and mechanical properties of the composites. The following revisions are needed before it is well suited for publication.

The language features a large number of minor flaws, typos and mistakes, which needs to be corrected considerably to meet the high language requirement of Materials. To list just a few: line 310-311, EDS measurements do not allow to estimate of the bright areas correspond to the solid solution or intermetallic phases identified by XRD.”

Answer:

The authors of the manuscript thank the reviewer for taking the time to review. The authors agree with the reviewer that the language features a large number of minor flaws, typos, and mistakes, which need to be corrected considerably to meet the high language requirement of Materials. In accordance with the reviewer's comments, the manuscript was linguistically checked. All changes made in the revised version of the manuscript are marked with red colour in the text.

For example, sentence: “EDS measurements do not allow to estimate of the bright areas correspond to the solid solution or intermetallic phases identified by XRD.” was changed at “EDS measurements do not allow to identify whether the analysed area is an intermetallic or solid state. Therefore previously presented XRD measurement was applied (Figure 9).”

“The introduction is not well organized. Currently there is only one paragraph. It needs to be re-paragraphed to make it more readable.

Please include a bit more introduction about the application background of ceramic-metal composites in the very beginning of the ‘Introduction’ section. e.g., these unusual properties (taking the advantage of the properties of individual material) have made ceramic-metal composites quite promising to be used in the field of biomedical (10.1016/j.jmbbm.2017.07.024), aerospace (Composites Part B 82 (2015) 13-22) and other engineering applications.”

Answer:

Authors agree with the Reviewer that the introduction is poorly written. Introduction was improved according the Reviewer suggestions. Authors added information about potential applications and the scientific background. Proper references were added, as well. Authors added the text in the manuscript:

“One of the fastest growing fields of mechanical and materials engineering is research on the fabrication of innovative ceramic materials. One of these materials are ceramic-metal matrix composites. Ceramic-metal composites belong to the groups of structural and functional engineering materials. Manufacturing such a material makes it possible to produce composites with different physical and chemical properties than a single-component material. This is particularly advantageous for the materials with applications as i.e. thermal barriers. Ceramics-metal composites are characterized by high corrosion resistance at high temperatures, thermal resistance, and high mechanical strength. The addition of a metallic phase into the ceramic matrix positively increases the fracture toughness. Although ceramic-metal composites have already been widely applied to various applications, the knowledge of technological processes of their forming and basics describing the relations between the structure and properties is still insufficient.”

“Taking the advantage of the properties of individual material (e.g. electric conductivity and ductility of metal, hardness and heat-resistance of ceramics) have made ceram-ic-metal composites very promising materials to be used in the field of biomedical [10], aerospace [11], electrotechnics [12] and other engineering applications [13,14].”

“The use of titanium and nickel as additional metallic components allowed the sintering process to produce intermetallic phases in the composite, which significantly affect the properties of the composite.”

“The determination of the dependence between the content of the composites components, microstructure (including the formation of intermetallic phases), and mechanical proper-ties of composites is a new contribution in the field of engineering manufacturing of new composite materials and the results of this study are presented in the paper.”

“The section ‘Materials and Methods’ can be subdivided (by adding subtitles) to make it more readable.”

Answer:

The authors agree with the Reviewer that the separation of the “Materials and Methods” will significantly increase the quality of the manuscript. The Authors divided the chapter into three sections (Base powders, Slurries, Composites). According to other Reviewer suggestion “Results and discussion” chapter is also divided. All changes made in the revised version of the manuscript are marked with red colour in the text.

“Fig. 1 has a rather low quality. They can hardly be seen clearly.”

Answer:

The authors agree with the reviewer that Fig. 1 has rather low quality. Therefore, in accordance with the reviewer's comments, Figure 1 has been changed.

“Add scale lines in the vertical axes in Fig. 3.”

Answer:

The authors agree with the reviewer that is necessary to add scale lines in the vertical axis in X-ray diffractograms of (A) alumina powder, (B) nickel powder, (C) titanium powder. Figure 3 has been corrected as suggested by the Reviewer.

“Fig. 4 has a rather low quality. The difference can hardly be seen clearly.”

Answer:

The authors agree with the Reviewer that Fig. 4 has rather low quality. Figure 3 has been modified.

“The vertical axes in Figs. 8-9 do not have any scale lines.”

Answer:

The authors agree with the reviewer that is necessary to add scale lines in the axes in Figs. 8-9. Figures 8-9 has been corrected as proposed by the reviewer.

“Fig. 12 has two A) and B), it is very consuming. If they are just the corresponding locally zoomed-in images, they need to be positioned properly with only one A) and B).”

Answer:

According to Reviewer suggestion Authors changed the description of the images for A, A’ for Series I and B, B’ for Series II. Images A and B corresponds to the areas with separated particles of Ni and Ti while A’ and B’ images reflects areas composed of both elements: titanium and nickel which are in the solid solution. Proper text was added in the manuscript.

“Normally the figure caption should be a phrase, i.e., ‘Fig. 7 The typical samples obtained by the slip casting method in the raw state’, where ‘were’ should be deleted.”

Answer:

Authors delete “were” in the sentence.

All changes made in the revised version of the manuscript are described above and marked with red color in the text. I again appreciate the help from the reviewers in improving the manuscript and hope the paper will be accepted for publication.

Yours sincerely,

Justyna Zygmuntowicz

Warsaw University of Technology

Faculty of Materials Science and Engineering

141 Woloska Str.

02-507 Warsaw

Poland

Round 2

Reviewer 1 Report

The authors revised their manuscript according to my suggestions. Thus the manuscript can be accepted for publication

Reviewer 2 Report

It can be accepted.

Reviewer 3 Report

Thank you for the response, I have no further comments.